# Effects of Arginine Supplementation on Athletic Performance Based on Energy Metabolism: A Systematic Review and Meta-Analysis

**DOI:** 10.3390/nu12051300

**Published:** 2020-05-02

**Authors:** Aitor Viribay, José Burgos, Julen Fernández-Landa, Jesús Seco-Calvo, Juan Mielgo-Ayuso

**Affiliations:** 1Glut4Science, Physiology, Nutrition and Sport, 01004 Vitoria-Gasteiz, Spain; aitor@glut4science.com; 2Department of Nursing and Physiotherapy, University of León, 24071 León, Spain; joseburgos88@hotmail.com (J.B.); julenfdl@hotmail.com (J.F.-L.); 3Institute of Biomedicine (IBIOMED), Physiotherapy Department, University of Leon, Researcher of Basque Country University, Campus de Vegazana, 24071 Leon, Spain; dr.seco.jesus@gmail.com; 4Department of Biochemistry Molecular Biology and Physiology, Faculty of Health Sciences, University of Valladolid, 42004 Soria, Spain

**Keywords:** aminoacids, ergogenic aids, physical performance, nitric oxide, aerobic, anaerobic

## Abstract

Nitric oxide related ergogenic aids such as arginine (Arg) have shown to impact positively on sport performance through several physiological and metabolic mechanisms. However, research results have shown to be controversial. The great differences regarding required metabolic pathways and physiological demands between aerobic and anaerobic sport disciplines could be the reasons. The aim of this systematic review and meta-analysis was to evaluate the effects of Arg supplementation on aerobic (≤VO_2_max) and anaerobic (>VO_2_max) performance. Likewise, to show the effective dose and timing of this supplementation. A structured search was carried out in accordance with PRISMA^®^ (Preferred Reporting Items for Systematic Reviews and Meta-Analyses) statement and PICOS guidelines in PubMed/MEDLINE, Web of Science (WOS), and Scopus databases from inception to January 2020. Eighteen studies were included which compare Arg supplementation with placebo in an identical situation and testing its effects on aerobic and anaerobic performance tests. Trials analyzing supplementation with other supplements were removed and there was not athlete’s level, gender, ethnicity, or age filters. The performed meta-analysis included 15 studies and random effects model and pooled standardized mean differences (SMD) were used according to Hedges’ g. Results revealed that Arg supplementation could improve aerobic (SMD, 0.84; 95% CI, 0.12 to 1.56; magnitude of SMD (MSMD), large; I2, 89%; *p* = 0.02) and anaerobic (SMD, 0.24; 95% CI, 0.05 to 0.43; MSMD, small; I2, 0%; *p* = 0.01) performance tests. In conclusion, acute Arg supplementation protocols to improve aerobic and anaerobic performance should be adjusted to 0.15 g/kg of body weight ingested between 60–90 min before. Moreover, chronic Arg supplementation should include 1.5–2 g/day for 4–7 weeks in order to improve aerobic performance, and 10–12 g/day for 8 weeks to enhance anaerobic performance.

## 1. Introduction

Athletes often turn to nutritional supplements in order to maintain health and maximize athletic performance [1]. Among them, proteins and amino acids represent the most consumed ergogenic aids, with a frequency of 35–40% [2]. However, the use of nutritional supplements with vasodilatory function are increasing considerably in the sport field, given that there is strong evidence that its intake has a positive effect on athletic performance [3,4,5]. In this sense, although nitrate and beetroot juice are the most studied vasodilatory supplements in this field [6,7], arginine (Arg) is an amino acid that has shown a vasodilator effect because it participates in the synthesis and bioavailability of nitric oxide (NO) [4,8]. For this reason, Arg supplementation has been used by athletes in order to obtain improvements in athletic performance [9,10,11,12].

Although Arg is a non-essential amino acid for adults because is absorbed through dietary proteins [13] and synthesized in the small intestine from proline, glutamate, and glutamine [8,14], some research has shown that supplementation could be beneficial to increase athletic performance [15]. The most relevant benefit of Arg is related to NO synthesis and its role as a cell signaling molecule with physiological relevant effects [16]. NO has shown increased blood flow and improved muscle contraction, gas exchange, oxygen kinetics, and mitochondrial biogenesis [6,17]. Otherwise, Arg has also been shown to stimulate the release of growth hormone (GH) [18,19], which helps to promote cell growth and regulate the mobilization of fuels in the body that contributes to increase muscle mass and hypertrophy [20,21,22]. Moreover, Arg supplementation has presented a reduction of ammonia, lactate, fatty acids, and fat oxidation levels after exercise [23,24]. Likewise, Arg has shown an increase in glycerol post-exercise, with improved carbohydrate oxidation and oxygen efficiency [25,26], considering these potential benefits in endurance sport performance. Thus, Arg has displayed effects on different physiological and metabolic pathways that could improve athletic performance in both, endurance or “aerobic” and high intensity or “anaerobic” athletic performance [15].

Athletic performance in endurance sports, in which efforts commonly last 5 min or more and requires equal and/or less intensity than VO_2_max, is related to the capacity of circulatory and respiratory systems to supply fuel and resynthesize adenosine triphosphate (ATP) by oxidative metabolism [27,28]. Therefore, endurance performance is determined by maximal oxygen uptake (VO_2_max), ventilatory thresholds, and energy efficiency or economy [28,29]. In this sense, different Arg supplementation protocols (<7 days or acute) and dosages (6–10 g/day) have been shown to improve several physiological parameters and performance outcomes, such as time to exhaustion, mean power output, and exercise capacity in moderate-submaximal intensities [10,11,30]. These results could be explained only due to improvements in blood flow and oxygen supply to the muscles, because it seemed that longer supplementation periods are needed to enhance mitochondrial respiration and oxidative phosphorylation through NO pathway [6]. However, other studies did not show any improvement on total 5 km running time and on cycloergometer incremental test performance in experienced runners and recreationally active men, respectively [31,32].

On the other hand, performance in high intensities or anaerobic sports, requires greater intensity than VO_2_max and depends on different metabolic pathways related to exercise duration [33]. In this sense, while high energetic phosphagen system (<6 s duration) is determinant for explosive disciplines, glycolysis represents the main energetic pathway for exercises between 1–5 min (with an increased contribution of oxidative phosphorylation proportionally with time), determined as high-intensity (~1 min) and intensive efforts (<5 min) [33,34,35,36]. Regarding anaerobic performance, chronic Arg supplementation (45–56 days) with low (2 g/day) and high (12 g/day) dosages could led to improve performance in one maximum repetition (1RM) bench press, Wingate test, and VO_2_max intensity test [9,20]. These positive effects related with strength could be explained because of Arg enhance GH-releasing hormone, suppresses the endogenous GH-inhibiting hormone and increases insulin-like growth factor 1 (IGF-1) [37,38]. Moreover, Arg plays an essential role in the synthesis of Creatine, main substrate for phosphagen system and anaerobic performance [8]. However, other authors did not find any benefits on muscle strength, maximum number of repetitions, and sprint power after ingesting 6 g/day of Arg in both acute and chronic protocols [39,40,41].

Although Arg supplementation could be effective on aerobic and high-intensity sport disciplines performance mediated by several effects, current evidence is controversial and confusing. In this sense, there are a few systematic reviews that analyze the effects of Arg supplementation on different physiological and metabolic mechanisms in older adults and disease patients [42,43]. Moreover, a short systematic review examined the connection between Arg and citrulline and sport performance [15]. However, to the best of the authors’ knowledge, there is not clearly and quantitatively analyzed information and distinction regarding the different involved effects in both aerobic and anaerobic performance capacities in the literature, and this may be necessary to better understand Arg supplementation reasons and protocols. Therefore, we proposed carrying out a systematic review and meta-analysis on the effects of Arg supplementation on exercise performance according to the main energy metabolism system used during exercise, with the main aim of analyzing current evidence and evaluating its impact on performance in both, aerobic and high-intensity or anaerobic, disciplines. In addition, this manuscript aims to show the effective doses and ideal moment of its intake.

## 2. Methods

### 2.1. Literature Search Strategies

This systematic review and meta-analysis was performed in accordance with PRISMA^®^ (Preferred Reporting Items for Systematic Reviews and Meta-Analyses) statement guidelines [44] and the PICOS model for the definition of the inclusion criteria: P (Population): “athletes”, I (Intervention): “impact of the Arg supplementation on sport performance”, C (Comparators): “same conditions with control or placebo”, O (Outcome): “sport performance”, and S (study design): “clinical trial” [45]. A systematic search of the scientific literature was performed to investigate the effect of Arg supplementation on sports performance. Studies were found by searching Web of Science (WOS), PubMed/MEDLINE, and Scopus from inception to 23rd January 2020, using the following Boolean search equation: “L-Arginine”[All Fields] OR “arginine”[All Fields] OR “AAKG”[All Fields] OR “arginine alpha-ketoglutarate”[All Fields] AND supplementation[All Fields] AND ((“sports”[MeSH Terms] OR “sports”[All Fields] OR “sport”[All Fields]) OR (“exercise”[MeSH Terms] OR “exercise”[All Fields])) AND (“endurance”[All Fields] OR “performance”[All Fields] OR “aerobic”[All Fields] OR “anaerobic”[All Fields] OR “strength”[All Fields]). Over this search equation, other articles in this field were included by the snowball strategy. To identify duplicates and any potential missing studies, all titles and abstracts from the search were cross-referenced. The abstracts and titles were screened for a subsequent full-text review.

### 2.2. Inclusion and Exclusion Criteria

The inclusion criteria applied in this systematic review and meta-analysis to choice studies were (1) well-designed experiments that included Arg supplementation; (2) identical experimental condition in the placebo or control group; (3) testing the effects of Arg supplementation on sports performance; (4) clinical trial; (5) clear information concerning the Arg supplementation administration (timing and dosage); (6) published in any language; (7) clear information about funding sources; and (8) absence of conflict of interests. On the other hand, this exclusion criteria were used regarding the experimental procedures of the investigation: (1) Arg supplementation was mixed with other supplements or was a multi-ingredient compound; and (2) participants had a previous injury or health problems. There were no filters applied to the athletes’ athletic level, age, gender, or ethnicity to increase the analytic power of the analysis.

### 2.3. Study Selection

Two authors independently screened and agreed upon the selected studies for eligibility (A.V. and J.M.-A.). Likewise, after the inclusion/exclusion criteria were applied to each study, data on study source (including authors and year of publication), sample size, characteristics of the participants (level, race, and gender), supplement administration (dose and timing), and final outcomes of the interventions were extracted independently by two same authors (A.V and J.M.-A.) using a spreadsheet (Microsoft Inc, Seattle, WA, USA). Then, possible disagreements were resolved through discussion until a consensus was reached, or by third-party adjudication (J.F.-L.). In this sense, the Cohen´s kappa coefficient, which indicate the interrater reliability, between authors was above 90 with a level of agreement “almost perfect” [46].

### 2.4. Outcome Measures

The literature was examined regarding the effects of Arg supplementation on sports performance using athletic performance outcome variables classified according to the duration of tests used. As 5 min test seems to be reliable in determining maximal aerobic velocity and therefore, how long a subject can maintain the lowest intensity at which VO_2_max was achieved [47,48], authors established this criteria to arrange analyzed performance outcomes variables in ≤VO_2_max and >VO_2_max. Two studies met both criteria and, hence, were included in both performance analysis. Concretely, the outcomes obtained by Repeated Sprint Ability Test (RSAT), strength exercises as isokinetic flexion, isokinetic extension, and bench press, 1 min all out test, 1 km time trial (TT) and Wingate Test were included in >VO_2_max. On the other hand, the outcomes obtained by Incremental test to exhaustion, 2 × 5 km TT, 60 min test at 80% of ventilatory threshold (VT), a 16.1 km TT and 2 × 6 min running followed by a running test until exhaustion. Harvard Step Test to measure VO_2_max. capacity were included in ≤VO_2_max. For the statistical analysis, the sample sizes, means and standard deviations of the different outcomes studied were extracted both in the group supplemented with Arg and in the control group and in the pre and post treatment. When there was no numerical data, it was requested to the authors or if the data were plotted as figures, the values were estimated based on the pixel count using images calibrated in ImageJ software (National Institutes of Health, Bethesda, MD, USA).

### 2.5. Publication Bias

Publication bias was assessed using Egger’s statistic test, where bias was deemed to be present at *p* = < 0.05 [49]. Corresponding funnel plots were created for visual interpretation, followed by an Egger’s statistic to confirm or refute publication bias (Figure 1). Egger’s analyses suggest that publication bias did not present finding in anaerobic performance (z = 0.786; *p* = 0.432). However, funnel plot showed publication bias in aerobic performance data (z = 2.873; *p* <0.05).

### 2.6. Quality Assessment of the Experiments

In accordance with the Cochrane Collaboration Guidelines [50], methodological quality and risk of bias were evaluated by two authors independently (A.V. and J.M.-A.), and disagreements were resolved by discussion and/or third-party author (J.F.-L.). The interrater reliability (Cohen’s kappa) was 90 with a level of agreement “almost perfect” [46]. The list was separated in six different domains: selection bias (random sequence generation, allocation concealment); performance bias (blinding of participants and researchers); detection bias (blinding of outcome assessment); attrition bias (incomplete outcome data); reporting bias (selective reporting); and other types of bias. The domains were considered as ‘low’ if criteria met a low risk of bias (probable bias unlikely to seriously alter the results) or ‘high’ if criteria presented a high risk of bias (probable bias that seriously weakens confidence in the results), or it was considered ‘unclear’ (plausible bias that raises some doubt about the results), if the risk of bias was unknown. Full details or each article and domains are presented in Figure 2 and Figure 3.

### 2.7. Statistical Analysis

Participants’ characteristics data are reported as mean ± standard deviation or mean ± error standard of the mean. Review Manager (Revman) version 5.3 (Copenhagen: The Nordic Cochrane Centre, The Cochrane Collaboration, 2014), was used to assess the quality of the experiments and to interpret them in risk of bias figures. Same software was utilized to make descriptive analyses and meta-analytic statistics. In order to compare the ingestion of Arg vs. placebo, the number of participants, the standardized mean difference (SMD), and the standard error of the SMD were performed for each trial. Hedges’ g [51] was used to calculate SMD of each study’s group. An overall effect and its 95% confidence interval (CI) were calculated weighting SMD by the inverse of the variance. In addition, both control and experimental group’s SMD were subtracted to obtain the net treatment effect and pooled SD of changes scores were used to calculate variance. Random effects model was used based on DerSimonian and Laird method [52]. The Cohen criteria were followed to interpret the magnitude of SMD (MSMD): <0.2, trivial; 0.2–0.5, small; 0.5–0.8, moderate; and >0.8, large [53].

Statistic I2 was calculated, as indicator of the percentage of observed total variation within studies due to real heterogeneity rather than chance, with the objective to avoid errors when using Q statistic in assessing heterogeneity [50]. I2 values are included from 0 to 100%, representing a small amount of inconsistency between 25% and 50%, medium amount of heterogeneity between 50% and 75%, and a large amount of heterogeneity when I2 value is higher than 75% [54]. In this sense, low, moderate, and high adjectives would be accepted referring to I2 values of 25%, 50%, and 75%, respectively, although a restrictive categorization would not be adequate in all circumstances [54,55,56].

## 3. Results

### 3.1. Main Search

The literature search through electronic database identified a total of 120 records according to the selected search equation and 11 additional studies were included through reference list searches. However, only 18 studies met all the inclusion criteria (Figure 4). From these initial 131 articles, 16 were remove after duplication analysis. Out of remaining 115 studies, 57 were excluded: 17 were reviews and 40 were not related with humans. Thus, 58 articles were full-text assessed for eligibility. From them, 40 were removed with reasons: 21 included more than one supplement, 7 used other supplements, 8 were performed in unrelated subjects, and 4 because of unsuitable outcomes. Thereby, a total of 18 were included in the analysis for this systematic review. However, due to insufficient data for carrying out the meta-analysis, three studies [25,57,58] were removed from quantitative synthesis and, thereby, a total of 15 were included for analysis.

### 3.2. Arginine Supplementation

Table 1 displays that eight studies followed a randomized, double-blind, placebo-controlled design [9,11,12,20,31,59,60,61], seven randomized, double-blind, cross-over, placebo-controlled design [10,25,39,40,57,58,62] and three randomized, placebo-controlled design [32,41,63]. None of the included studies presented any conflict of interests. Out of 15, 6 studies were carried out in endurance or aerobically trained healthy athletes [11,25,31,40,59,61], 3 in team sport athletes (soccer) [9,41,60], 2 in combat sports athletes (judo and wrestlers) [10,58], 1 in resistance-trained males [20] and remaining 6 in healthy, recreationally active males and females [12,32,39,57,62,63]. Total study participants included in the systematic review were 394 (386 males and 8 females). Out of them, 282 were trained or elite athletes. Sixteen studies divided participants in placebo (PLA) or control (CON) group and Arginine (ARG) group [9,10,20,25,31,32,39,40,41,57,58,59,60,61,62,63]. The remaining two assigned participants in PLA or CON and two ARG groups [11,12], in order to compare different doses.

Supplementation was ingested by participants in both acute and chronic protocols (Table 1). Eleven studies carried out an acute Arg ingestion, between 30 min and 2 days before tests. Concretely, 5 used Arg supplementation 60 min before test [10,25,57,58,60], 2 studies 90 min before [11,62], 1 study [59] 80 min before, 1 study [39] 4 h and 30 min before test, another one [40] provided supplementation 60 min and 30 min before test, and a last one [63] 30 min previously. Remaining 7 studies used chronic supplementation protocols between 7 and 45 days: in 3 of them, participants ingest Arg during 4 weeks or 28 days [12,31,61], while 1 study provided supplementation during 7 days [32], Mor et al. [41] used a 14 days protocol, Pahlavani et al. [9] used a longer protocol of 45 days and Campbell et al. [20] during 8 weeks or 56 days.

In 14 studies, the Arg supplementation dose was absolute. Out of them, in 8 studies the protocol consisted of 6 g/day of Arg [31,32,40,41,58,59,62,63], in 2 trials 2 g/day was used [9,57], in 1 study 12 g/day was ingested [20], another one provided 3.7 g [39], Abel et al. [61] used 5.7 g/day and 2.85 g/day for each group and the remaining one provided 1.5 g/d and 3 g/day [12]. Relative supplementation dose was used in 3 studies: 0.075 g∙kg^−1^ body mass [25] and 0.15 g∙kg^−1^ body mass [10,60]. The exception was the trial carried out by Hurst et al. [11], in which both, absolute (6 g/d) and relative (0.15 g∙kg^−1^ body mass), were provided to participants.

### 3.3. Effect of Arginine on Anaerobic Performance (>VO_2_max)

Table 2 presents the different tests carried out in selected studies determining performance above VO_2_max intensity and measured outcomes. RSAT was used in 2 studies [41,60]. Strength exercises as isokinetic flexion, isokinetic extension, and bench press were used in other 2 studies [40,59], 1 min all out test was carried out by Bailey et al. [32], 1 km TT by Hurst et al. [11], and Wingate Test in Olek et al. [57] and Campbell et al. [20] trials.

The supplementation of Arg showed improvements in Wingate Test and Upper Body 1RM [20] and in chin-ups total repetitions [39] but the remaining 8 trials did not find any differences in treatment [11,32,40,41,57,58,59,60].

### 3.4. Effect of Arginine on Aerobic Performance (≤VO_2_max)

Table 3 shows the tests, measured outcomes and effects of Arg supplementation on aerobic or ≤VO_2_max performance. Out of 11 studies, 6 tested performance with a cycloergometer [10,11,12,25,32,61], 4 in running protocols [20,31,62,63], and 1 remaining step protocol [9]. Incremental test to exhaustion was used in 5 trials [10,12,20,32,61] and 2 × 5 km TT in 2 other studies [31,63]. Forbes et al. [25] carried out a 60 min test at 80% of VT, Hurst et al. [11] a 16.1 km TT and Vanhatalo et al. [62] 2 × 6 min running followed by a running test until exhaustion. Lastly, Pahlavani et al. [9] used Harvard Step Test to measure VO_2_max capacity.

Improvements in performance were found in four studies, corresponding to reduced time to exhaustion [10], increased power output in one of the two ARG group [11], physical working capacity [12] and performance score [9]. In the remaining trials [11,20,25,32,59,61,62,63], treatment did not show any performance improvements.

### 3.5. Effect on Anaerobic Performance (>VO_2_max) Meta-Analysis

Figure 5 shows that Arg presented small and significant effect on anaerobic performance (Hedges’ g (SMD) = 0.24; 95% CI (0.05–0.43); *p* = 0.01). The meta-analysis reported small amount of inconsistency between studies reviewed (I^2^ = 0%; *p* = 0.85). Three analyzed studies presented positive effects on anaerobic performance outcomes after Arg supplementation. Among them, Campbell et al. [20] obtained strong improvements in upper body 1RM and in Wingate Test parameters like peak power, time to peak power and rate of fatigue. On the other hand, Hurst et al. [11] showed small and not significantly positive effects on time to completion and power output during 1 km TT in cycloergometer, as well as Birol et al. [60] regarding total sprint time during RSAT. Although these results were not significant, clear Arg favorable tendencies were shown in original trials. In this sense, due to statistical circumstances, such as low participants number and followed criteria, results could not be considered as statistically significant (*p* < 0.05) in each study. However, in our meta-analysis, once all studies were analyzed, significantly positive results were obtained in favor of Arg supplementation, showing that, on the whole, Arg effects regarding anaerobic outcomes were favorable (*p* = 0.01).

### 3.6. Effect on Aerobic Performance (≤VO_2_max) Meta-Analysis

Arg showed large and significant effect on aerobic performance (Hedges’ g (SMD) = 0.84; 95% CI (0.12–1.56); *p* = 0.02) (Figure 6). The meta-analysis reported a large heterogeneity between the studies reviewed (I^2^ = 89%; *p* < 0.001). Regarding the effects, Camic et al. [12] and Pahlavani et al. [9] showed large improvements in favor of Arg supplementation in physical working capacity at the fatigue threshold, and in the Harvard Step Test performance score, respectively. Moderate positive results were obtained by Yavuz et al. [10] during a cycloergometer incremental test to exhaustion. Similarly, Hurst et al. [11] showed moderate improvements in power output during 16.1 km TT in cycloergometer. However, these results were only observed in one of the two groups analyzed, with no small significant positive results in time to completion for group 1 and 2, and power output for group 1.

## 4. Discussion

The main objective of this systematic review and meta-analysis was to analyze and summarize the current evidence around the effects of Arg supplementation on aerobic and high-intensity anaerobic athletic performance. Results obtained mainly indicate that Arg specific supplementation protocols led to significant improvements both on anaerobic (>VO_2_max) and aerobic (≤VO_2_max) performance. Regarding dosage, differences are observed between acute and chronic protocols, suggesting that both strategies could have different mechanisms and responses. In that line, 0.15 g/kg body mass (≈10–11 g) of Arg supplementation ingested between 60–90 min before exercise (acute protocol) have shown to improve on anaerobic and aerobic performance outcomes. In addition, 1.5–2 g/day Arg supplementation for 4–7 weeks (chronic protocol) and longer doses (10–12 g/day for 8 weeks) may also be beneficial to enhance aerobic and anaerobic performance, respectively.

Regarding the well-known role of Arg related with NO production and increased blood flow and vasodilation, it has been suggested to favorably impact on training adaptations [64]. However, there are few studies analyzing these effects. In fact, due to the limited and controversial results about Arg supplementation on performance variables, Arg supplementation is not included as an ergogenic aid with strong evidence [4,5]. In this line, although 6 g/day seems to be the most used protocol, there are no clear timing and dose protocols related with sport performance. While, results obtained in this systematic review and meta-analysis indicated that an Arg supplementation with 6 g/day did not have significative effects on aerobic and anaerobic performance outcomes, higher doses (≈10 g) seem to be more effective on both anaerobic and aerobic performance outcomes. In this sense, these results depend on the timing of Arg ingestion. Given that nitrate and nitrite blood levels are increased 2.5–3 h after nitrate-rich supplement intake [65], and Arg pharmacokinetic tests have shown a maximum plasma concentration between 30–90 min after ingestion [66], acute ingestion protocols of Arg and other supplements related with NO metabolism have been analyzed in several studies. However, chronic supplementation protocols seem to be necessary to obtain major adaptations in other NO supplements as beetroot juice or nitrate [6]. Present work suggests that when acute protocols are used, higher doses (≈10 g) are needed to show improvements on aerobic performance, while when chronic ingestion protocols are followed, lower doses (1.5–3 g/day) may be enough to show benefits. Therefore, it suggests that studying differences between acute and chronic supplementation protocols could be essential to understand what dosage is more adequate in order to obtain performance optimization.

The physiological and metabolic differences between aerobic and high-intensity sport disciplines are relevant [33]. While anaerobic performance depends on the capacity to rapidly synthesize ATP mainly via phosphagen system or cytosolic glycolysis (overreaching the capacity to oxidize lactate), performance in aerobic sports like road cycling or endurance running is determined by the capacity to oxidize lactate and supply ATP demands via oxidative phosphorylation [28,35,36]. In this line, nutrition and supplementation represents an important tool to optimize sport performance, especially in aerobic disciplines [5,67]. Given that NO pathway is involved in some beneficial physiological aerobic mechanisms, such as improved economy, gross efficiency, ventilatory kinetics, and metabolic responses, trying to optimize them with supplementation could be an adequate strategy [68]. Arg plays an essential role on endogenous NO synthesis and, thus, on some major physiological responses. Therefore, Arg supplementation could lead to improve training and competing performance due to the higher capacity to maintain work while prolonged exercising [9,10,12]. Because of that, Arg supplementation could be recommended for the improvement of anaerobic and aerobic sport disciplines performances.

### 4.1. Effect on Anaerobic Performance (>VO_2_max)

Explosive (6 s), high-intensity (<1 min) and intensive (<5 min) efforts are determined by the anaerobic power and capacity, but there are some differences on metabolic and physiological demands between them [33,35]. From metabolic point of view, the main fuel substrates to rapidly synthesize ATP in anaerobic exercise is phosphocreatine and lactate. Therefore, intramuscular creatine content, glycolytic power, and buffer capacity are determinant factors [35,36]. On the other hand, physiologically, neuromuscular mechanisms, muscle contraction function, and structural factors such as muscle fiber composition (type II) are essential to ensure high intensity performance [36]. In this line, NO-pathway related supplements have shown, among other effects, to have specific impact on type II fibers improving anaerobic performance [69]. Moreover, Arg supplementation may have direct effects improving creatine resynthesis and reducing lactate accumulation [23,38].

Regarding anaerobic performance, the most commonly used measurement tests are Wingate test, short time trials (1–2 km), isokinetic exercises, 1RM and, in team sports, the RSAT [70,71,72]. In this sense, high doses (12 g/d) of chronic Arg supplementation (8 weeks) led to significant improvements in Wingate test parameters (peak power, time to peak power, and rate of fatigue), as well as enhanced 1RM performance in well resistance-trained subjects [20]. However, although they did not observe significant effects on RSAT total sprint time [60] and time to complete and average power output during a 1 km TT [11], high doses of Arg (0.15 g/kg ≈ 10–11 g) ingested 60–90 min before exercise led to a clear improvement tendency in these anaerobic parameters. In the same line, 2–6 g/d of acute Arg supplementation (60–90 min before exercise) did not show significant performance improvements in Wingate Test, 20” all out sprint and 1 km TT, although tendency in this last trial was also positive [11,57,58]. Likewise, 6 g/d for 7–14 days did not show any significant improvements in 1 min all out cycling sprint in recreationally active men [32] and in anaerobic sprint capacity (6 consecutive sprints) in soccer players [41]. Moreover, 3.7–6 g/d of Arg supplementation did not display significant beneficial effects regarding acute (30–80 min) or chronic (8 weeks) protocols on strength performance and sprinting capacity. Although it could be reasonable to expect greater anaerobic improvements with Arg supplementation due to enhanced blood flow and reduced lactate accumulation, among other mechanisms, positive results obtained in trials showed small significant effects but these had a tendency that was clear in some of analyzed trials. However, when global analysis was made, it showed that these improvements were represented with a higher and significant statistical power.

Therefore, these results suggest that regardless of fitness level and timing of supplementation, lower dosages are insufficient to improve high intensity anaerobic performance and that higher doses could be needed in both acute and chronic protocols. Regarding this, a supplementation of 0.15 g/kg body weight taken 60–90 min before exercise (acute protocol) or 10–12 g Arg supplementation for 8 weeks (chronic protocol) could led to improvements on anaerobic performance.

### 4.2. Effect on Aerobic Performance (≤VO_2_max)

Aerobic performance requires a great metabolic flexibility and a great cardiovascular capacity that supplies oxygen needs during prolonged exercise [28,73]. Although genetic predisposition represents an important requirement to perform in aerobic disciplines, looking for training and nutrition protocols to improve performance in such demanding disciplines is essential [74]. Beyond nutritional and hydration aids like carbohydrate-rich foods and electrolyte drinks, ergogenic aids such as caffeine or vasodilatory aids are often used [1,3]. In this regard, 1.5–3 g/d of chronic Arg supplementation (4–7 weeks) improved physical working capacity at the fatigue threshold in amateur and healthy-untrained subjects [9,12], showing lower fatigue related metabolites concentration such as ammonia and lactate, as well as a higher muscular power, endurance, and blood flow. On the other hand, a range of 2.85–5.7 g/d Arg supplementation for 4–8 weeks did not show statistical effects on time to exhaustion in running and cycling in endurance well-trained athletes [31,61]. Even higher dosage of 12 g/d Arg supplementation during more prolonged duration (8 weeks), did not present any improvements on time to exhaustion while running an incremental test protocol in resistance trained athletes [20]. These controversial results could be explained attending to subjects physical fitness, considering that Arg supplementation could have lower positive effects in well-trained subjects comparing with untrained ones as documented in other vasodilatory supplements [75].

Regarding acute supplementation, 0.15 g/kg (≈10–11 g) of Arg ingested between 60–90 min before exercise, led to improve time to exhaustion and power output in cycling endurance test in both trained and untrained subjects [10,11]. These effects could be explained due to reduced VO_2_ slow components, anaerobic reserves use, reduced metabolites of fatigue, and ATP cost of muscle force production [10,11]. However, Arg supplementation within 5.7–6 g between 30–90 min previous to exercise, seemed not to show beneficial effects on total running time and average power output at ventilatory threshold in both endurance-trained and recreationally active subjects [11,25,62,63]. These differences suggest that when it comes to improving the physiological and metabolic responses related with enhanced aerobic performance, acute Ag supplementation protocols with higher doses (≈10–11 g absolute or 0.15 g/kg), independently of fitness status, should be followed.

Chronic and acute supplementation protocols could have different physiological mechanisms of action and these may be closely and directly related with different Arg dosages. In this regard, results obtained in this systematic review and meta-analysis demonstrated that aerobic performance could be optimized when Arg supplementation is followed in both acute (higher dosage) and chronic (lower dosage) protocols. However, these effects have to be taken with caution due to the publication bias found in the included aerobic performance included studies. Finally, considering that different physiological responses are documented on trained and untrained athletes when other NO-related supplement (beetroot juice) is ingested [6], it suggests that Arg supplementation dosages may need to be adjusted according to the subjects fitness level.

### 4.3. Strength, Limitations, and Future Lines of Research

The few total number of studies (n = 18) carried out in this systematic review and meta-analysis related to Arg supplementation could represent a limitation. On the other hand, the exhaustive methodology followed by authors in both systematic review and meta-analysis, concerning studies selection and outcomes analysis, interpretation, and quantification represents also a strong point. In fact, when assessing the review and meta-analysis quality by AMSTAR guidelines, the score obtained was ‘high quality’ [76]. In the same line, this meta-analysis was carried out with a strong statistical power and confidence, which constitute an important point in favor. Regarding supplementation, all included studies use only Arg with no other supplements, compared with a placebo, what limits the possibility to obtain synergistic or antagonistic effects along with other compounds. Moreover, there are some other limitations regarding supplementation protocols, doses and timing. Due to controversial results around Arg supplementation and its effects in the current literature [5], adequate dosages and supplementation duration are not yet well established. Because of that, there were found out some different and heterogeneous supplementation and timing protocols that limited the extraction of strong conclusions. In addition, performance outcomes and protocols used in studies were dissimilar and, beyond their classification into anaerobic and aerobic performance, differences were found within each section regarding mainly used metabolic pathways. In the same line, the heterogeneity and the publications bias obtained in aerobic performance studies, mean that practitioners should take these results with caution when applying the analyzed outcomes to reality.

Future lines of research should focus on homogenizing Arg supplementations protocols regarding dosages and timing, due to the current miscellaneous methods. To our understanding, and taking into account the results obtained, new Arg dosages should be proposed in favor of higher and chronic supplementation protocols, as well as studying the differences between elite and amateur subjects. Finally, a novel research line should be considered studying the Arg absorption capacity and involved mechanisms in athletes, as this could represent a crucial understanding of supplementation knowledge.

## 5. Conclusions

This systematic review and meta-analysis have shown that Arg supplementation could have positive effects on anaerobic and aerobic performance related physical test outcomes. The effective dose of Arg supplementation in acute protocols should be adjusted to 0.15 g/kg (≈10–11 g) ingested between 60–90 min before exercise for improve both aerobic and anaerobic disciplines. On the other hand, chronic Arg supplementation of 1.5–2 g/day for 4–7 or longer doses (10–12 g/day for 8 weeks) presented a positive impact on aerobic and anaerobic performance, respectively.

## Figures and Tables

**Figure 1 nutrients-12-01300-f001:**
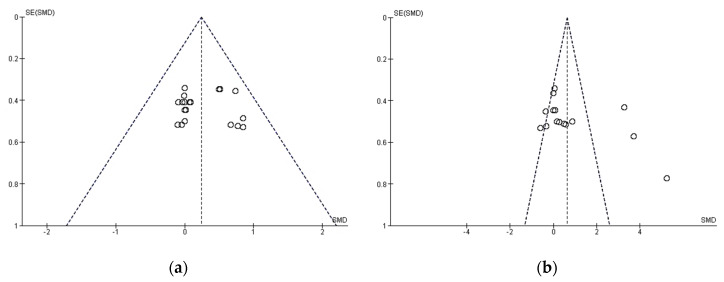
Funnel plot of standard error of anaerobic (**a**) and aerobic; (**b**) performance data by Hedges’ g. SE: standard error; SMD: standardized mean difference.

**Figure 2 nutrients-12-01300-f002:** Summary of risk of bias: authors’ judgements about each risk of bias item for all included studies. 
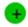
 indicate low risk of bias; 
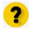
 indicate unknown risk of bias; 
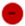
 indicate high risk of bias.

**Figure 3 nutrients-12-01300-f003:**
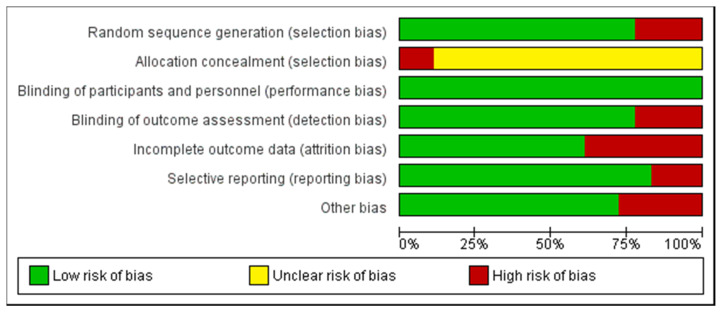
Graph of risk of bias: authors’ judgements about each risk of bias item presented as percentages across all included studies.

**Figure 4 nutrients-12-01300-f004:**
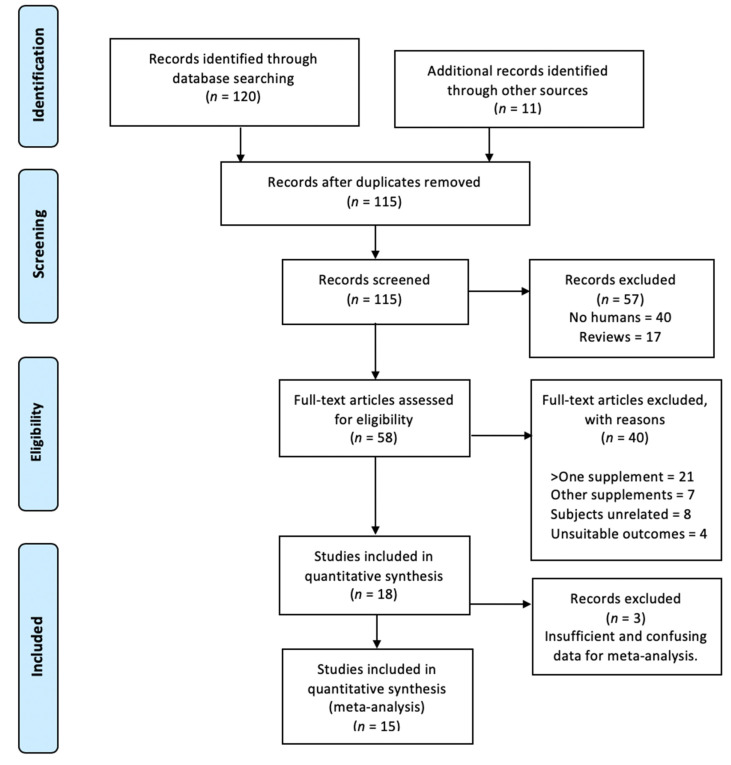
Preferred Reporting Items for Systematic Review and Meta-Analyses (PRISMA) flow diagram.

**Figure 5 nutrients-12-01300-f005:**
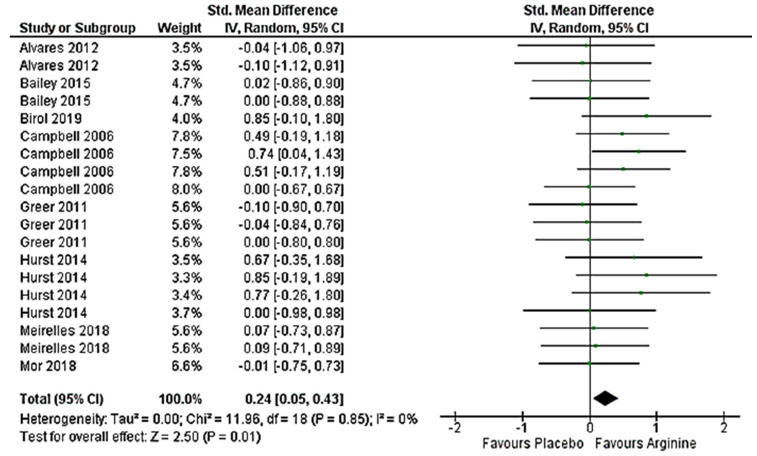
Forest plot comparing the effects of arginine supplementation on anaerobic (>VO_2_max) performance.

**Figure 6 nutrients-12-01300-f006:**
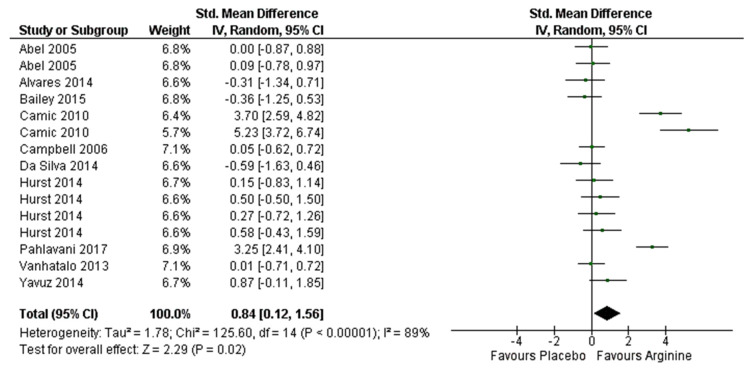
Forest plot comparing the effects of arginine supplementation on aerobic (≤VO_2_max) performance.

**Table 1 nutrients-12-01300-t001:** Participant and intervention characteristics of the studies included in the systematic review and meta-analysis

General Characteristic	Intervention Characteristic		Studies Included
Study Design	Randomized, double-blind and placebo-controlled		8 studies [9,11,12,20,31,59,60,61]
	Randomized, double-blind, cross-over and placebo-controlled		7 studies [10,25,39,40,57,58,62]
	Randomized and placebo-controlled		3 studies [32,41,63]
Conflict of Interests	None		18 studies [9,10,11,12,20,25,31,32,39,40,41,57,58,59,60,61,62,63]
Subjects Characteristics	Endurance-trained		6 studies [11,25,31,40,59,61]
	Fight sports athletes		Judo—1 study [58]
			Wrestlers—1 study [10]
	Soccer athletes		3 studies [9,41,60]
	Resistance-trained		1 study [20]
	Active		6 studies [12,32,39,57,62,63]
Type of Arginine Supplement	L-Arginine		15 studies [9,10,11,12,25,31,32,40,41,57,58,59,60,62,63]
	Arginine Aspartate		1 study [61]
	Arginine Alpha-Ketoglutarate		2 studies [20,39]
Type of Arginine Administration	Absolute		14 studies [9,12,20,31,32,39,40,41,57,58,59,61,62,63]
	Based on individual’s body mass		3 studies [3,10,13]
	Both		1 study [4]
Dose Used	12 g/day		1 study [20]
	6 g/day		8 studies [31,32,40,41,58,59,62,63]
	3.7 g/day		1 study [39]
	2 g/day		2 studies [9,57]
	5.7 d/day (group 1) and 2.85 g/day (group 2)		1 study [61]
	1.5 d/day (group 1) and 3 g/day (group 2)		1 study [12]
	0.075 g·kg^−1^ body mass		1 study [25]
	0.15 g·kg^−1^ body mass		2 studies [10,60]
	6 g/day (group 1) and 0.15 g·kg^−1^ body mass (group 2)		1 study [11]
Time of Ingestion	Acute	60 min before test	5 studies [10,25,57,58,60]
		90 min before test	2 studies [11,62]
		80 min before test	1 study [59]
		4 h + 30 min before test	1 study [39]
		60 min + 30 min before test	1 study [40]
		30 min before test	1 study [63]
	Chronic	56 days or 8 weeks	1 study [20]
		45 days	1 study [9]
		28 days or 4 weeks	3 studies [12,31,61]
		14 days	1 study [41]
		7 days	1 study [32]

**Table 2 nutrients-12-01300-t002:** Summary of the studies included in the systematic review that investigated the effect of L-Arginine on >VO_2_max Performance (test lasting less than 5 min).

Author/s	Population	Intervention	Test	Outcomes	Main Conclusion
Alvares, T.S. et al., 2012	15 healthy male volunteers with previous resistance training experience. Arg group 26.3 ± 4.9 years vs. PLA 24.7 ± 1.8 years.	Randomized, double-blind, placebo-controlled. 6 g/ of L-Arg (80 min before test).	Dominant elbow flexion and extension exercise with an isokinetic dynamometer. 3 sets of 10 maximal voluntary contractions.	• Peak Torque	↔
				• Total Work	↔
Bailey S.J., et al., 2015	10 healthy, recreationally active men (19 ± 1 years).	Randomized, double-blind, placebo-controlled. 6 g/d of L-Arg (7 days).	Day 6: 1 min all out cycle sprint.	• Peak Power	↔
				• Total Work	↔
Birol A. et al., 2019	20 volunteer healthy male football players (18.30 ± 0.48 years).	Randomized, double-blind, placebo-controlled. L-Arg 0.15 g/kg/day (60 min before test).	RSAT: 12 × 20 m with 30 s rest.	• Total sprint time	↔
Campbell B., et al., 2006	35 resistance-trained adult men (39.8 ± 5.8 years).	Randomized, double-blind, placebo-controlled. 12 g/d (4 g × 3) Arg Alpha-Ketoglutarate (8 weeks).	Upper body flat bench 1RM.Wingate test Isokinetic leg extension 50 rep.	• Upper body 1RM	↑
				• Peak power	↑
				• Time to Peak power	↑
				• Rate to Fatigue	↑
				• Isokinetic leg extension	↔
Greer B.K. et al., 2011.	12 trained college-aged men (22.6 ± 3.9 years).	Randomized, double-blind, placebo-controlled, cross-over. 3.7 g Arg Alpha-Ketoglutarate (4 h + 30 min pre-test).	3 sets of chin-ups, reverse chin-ups and push-ups to exhaustion.	• Chin-ups	↑
				• Reverse Chin-ups	↔
				• Push-ups	↔
Hurst H.T., et al., 2014.	8 healthy, trained male cyclists (21.00 ± 1.41 years).	Randomized, double-blind, placebo-controlled. Group 1: 6 g L-Arg, Group 2: 0.15 g·kg^−1^ body mass (90 min before test).	1 km TT in cycloergometer.	• Time to complete Group 1	↔
				• Power output Group 1	↔
				• Time to complete Group 2	↔
				• Power output Group 2	↔
Liu T.H., et al., 2009.	10 elite male college judo athletes (20.2 ± 0.6 years)	Randomized, cross-over, placebo-controlled. 6 g/d L-Arg (2 days, 60 min before test).	13 × All out test: 20 s with 15 s rest. Cycloergometer.	• Total power	↔
Meirelles C.M., et al., 2018	12 healthy university students, resistance trained males. (27 ± 3 years).	Randomized, double-blind, cross-over, placebo-controlled. 6 g L-Arg (3 g 60 min before test + 3 g 30 min before test).	Bench press in a Smith Machine and unilateral knee extension of the right leg.	• Bench press repetitions	↔
				• Knee extension repetitions	↔
Mor A., et al., 2018	28 amateur male soccer players (18–30 years).	Randomized, placebo-controlled. 6 g/d L-Arg (14 days).	Running Anaerobic Sprint Test (RAST): 6 × 32 m with 10 s rest.	• Mean power	↔
Olek R.A., et al., 2010.	6 healthy, active, but not highly trained volunteers (23.2 ± 0.5 year).	Randomized, double-blind, cross-over, placebo-controlled. 2 g L-Arg (60 min before test).	3 × All out 30 s Wingate Tet in Cycloergometer with 4 min rest.	• Power output	↔

↑: statistically higher effects; ↓: statistically lower effects. ↔ No effect. 1RM: one maximum repetition; Arg: Arginine; PLA: Placebo; RSAT: Repeated Sprint Ability Test.

**Table 3 nutrients-12-01300-t003:** Summary of the studies included in the systematic review that investigated the effect of L-Arginine on ≤VO_2_max performance (test lasting 5 min or more).

Author/s	Population	Intervention	Test	Outcomes	Main Conclusion
Abel, T. et al., 2005	30 male endurance-trained athletes (Group 1: 38.5 ± 10 years; Group 2: 34.4 ± 8.6 years)	Randomized, double-blind, placebo-controlled. Group 1: 5.7 g/d Arg-Aspartate; Group 2: 2.85 g/d Arg-Aspartate. (4 weeks).	Incremental cycloergometer test	Time to exhaustion Group1	↔
				Time to exhaustion Group 2	↔
Alvares, T.S. et al., 2014	15 healthy experienced runners (11 males and 4 females) (36.8 ± 7.1 years).	Randomized, double-blind, placebo-controlled. 6 g/d of encapsulated L-Arg hydrochloride (4 weeks).	2 × 5 Km TT running with 10 min recovery	Total running time.	↔
Bailey S.J., et al., 2015	10 healthy, recreationally active men (19 ± 1 years).	Randomized, double-blind, placebo-controlled. 6 g/d of L-Arg (7 days).	Day 7: Time to exhaustion test in cycloergometer	Time to exhaustion.	↔
Camic, C.L. et al., 2010	50 college-aged men (23.9 ± 3.0 years).	Randomized, double-blind, placebo-controlled.3 groups: (a) placebo (n = 19); (b) 1.5 g/d Arg (n = 14); or (c) 3.0 g/d Arg (n = 17) (4 weeks).	Incremental test to exhaustion in cycloergometer	PWC_FT_ Group 1	↑
				PWC_FT_ Group 2	↑
Campbell B., et al., 2006	35 resistance-trained adult men (39.8 ± 5.8 years).	Randomized, double-blind, placebo-controlled. 12 g/d (4 g × 3) Arg Alpa-Ketoglutarate (8 weeks).	Bruce protocol: Incremental test running	Time to exhaustion	↔
Da Silva D.V., 2014	15 physically active and healthy volunteers (11 males and 4 females). Arg group: 36.8 ± 7.1 years; PLA: 30.6 ± 9.5 years).	Randomized, placebo-controlled. 6 g/d of L-Arg (30 min before test).	2 × 5 km TT running with 10 min recovery	Total running time.	↔
Forbes S.C., et al., 2013	15 aerobically trained men (age: 28 ± 5 years)	Randomized, double-blind, placebo-controlled, cross-over. L-Arg 0.075 g·kg^−1^ body mass (60 min before test).	60 min at 80% of VT incycloergometer	Average power output at ventilatory threshold.	↔
Hurst H.T., et al., 2014	8 healthy, trained male cyclists (21.00 ± 1.41 years).	Randomized, double-blind, placebo-controlled. Group 1: 6 g L-Arg, Group 2: 0.15 g·kg^−1^ body mass (90 min before test).	16.1 km TT in cycloergometer	Time to complete Group 1	↔
				Power output Group 1	↔
				Time to complete Group 2	↔
				Power output Group 2	↑
Pahlavani N., et al., 2017	56 male soccer players (20.85 ± 4.29 years).	Randomized, double-blind, placebo-controlled. 2 g/d L-Arg (45 days).	Harvard Step Test	Performance Score	↑
Vanhatalo A., et al., 2013	18 healthy, recreationally active male students (22 ± 3 year).	Randomized, double-blind, cross-over placebo-controlled. 6 g/d of L-Arg (90 min before test).	2 × 6 min running moderate test + 1 × running test until exhaustion	Time to exhaustion	↔
Yavuz, H.U., et al., 2014	9 volunteer elite male wrestlers (24.7 ± 3.8 years)	Randomized, placebo-controlled, cross-over. L-Arg 0.15 g·kg^−1^ body mass (60 min before test).	Incremental test to exhaustion in cycloergometer	Time to exhaustion	↑

↑: statistically higher effects; ↓: statistically lower effects. ↔ No effect. Arg: Arginine; PLA: Placebo; TT: Total Time trial; PWCFT: Physical working capacity at the fatigue threshold; VT: ventilatory threshold.

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
