# Peer review of "Effects of Arginine Supplementation on Athletic Performance Based on Energy Metabolism: A Systematic Review and Meta-Analysis"

_nutrients, 2020, doi:10.3390/nu12051300_

Round 1
Reviewer 1 Report
Effects of Arginine Supplementation on Athletic Performance based on energy metabolism: A Systematic Review and Meta-Analysis
The authors of the submitted manuscript conducted a systematic review and meta-analysis of studies to determine the effects of arginine supplementation on aerobic and anaerobic performance via metabolic energy pathways. A second aim of the submitted manuscript was to attempt at identifying the ideal dose and moment of intake of the supplement. The authors were thorough in their conduct of the systematic search of the literature and were able to identify a total of 15 studies fulfilling their inclusion/exclusion criteria. Because many of the included studies were of varying methodological design, the authors analyzed data to: a) determine whether long-term consumption of Arg is beneficial; b) determine the most effective time period for taking Arg before a workout; and, c) determine what would be the most effective Arg dosage. After carefully reviewing the findings from the included studies, the authors concluded that long-term Arg supplementation showed a positive outcome in both aerobic and anaerobic performance, which was related to physical test outcomes. The authors also learned that the best time period to take an Arg supplement was 60-90 min before exercise, and the most effective dosage corresponded with the athlete’s body weight (0.15 g/kg).
Although the authors should be commended for their work, this reviewer has some questions, comments, and points of clarification regarding the reporting of this manuscript that need to be addressed before considering the manuscript further for publication consideration. To help guide and direct the author’s efforts, the reviewer has raised each question, point of clarification or concern point-by-point below.
Methods:
- Section 2.3: It seems that only one rater screened and selected the studies for inclusion. According to the AMSTAR 2 checklist, for best methodological quality, systematic reviews and meta-analyses should include at least two screeners who should independently screen and agree upon the selected studies for eligibility. Alternatively, the two reviewers should each select a sample of the same eligible studies and achieve at least 80% agreement between those studies with the remainder of the studies being screened by one reviewer. This reviewer therefore requests that the authors please clarify whether or not the reviewers of the submitted manuscript adhered to this protocol or used a different protocol.
- Section 2.4: Please include a more in-depth description of the steps taken to perform the statistical analysis of the performance outcome variables, specifically VO2 max.
- Please provide a more comprehensive discussion of the methods used to code the 15 articles included in the analysis and how inter-rater reliability and validity were established with Kappa and Pearson’s r statistical procedures.
Results:
- Section 3.5/3.6: Please include a calculation of the Hedges’ g effect size to improve clarity of data presented. Please also provide the established reliability statistics.
- Please update and provide a clearer description for table 3.
Other comments:
- Please consider revising the submitted manuscript with AMSTAR 2 guidelines in consideration of improving the overall clarity and transparency of the manuscript with regard to the reported study findings.
- Please include a description of any publication bias procedures that were performed for the current study and state whether publication bias was existent/non-existent..
- Please double check and correct any typographical errors throughout the manuscript.
Author Response
Point-by-Point Response to Reviewer’s Comments
We would like to sincerely thank the reviewers for their helpful recommendations. We have seriously considered all the comments and carefully revised the manuscript accordingly. Revisions are highlighted in yellow through the manuscript to indicate where changes have taken place. We feel that the quality of the manuscript has been significantly improved with these modifications and improvements based on the reviewers’ suggestions and comments. We hope our revision will lead to an acceptance of our manuscript for publication in Nutrients.
In advance,
King regards
Reviewer 1
The authors of the submitted manuscript conducted a systematic review and meta-analysis of studies to determine the effects of arginine supplementation on aerobic and anaerobic performance via metabolic energy pathways. A second aim of the submitted manuscript was to attempt at identifying the ideal dose and moment of intake of the supplement. The authors were thorough in their conduct of the systematic search of the literature and were able to identify a total of 15 studies fulfilling their inclusion/exclusion criteria. Because many of the included studies were of varying methodological design, the authors analyzed data to: a) determine whether long-term consumption of Arg is beneficial; b) determine the most effective time period for taking Arg before a workout; and, c) determine what would be the most effective Arg dosage. After carefully reviewing the findings from the included studies, the authors concluded that long-term Arg supplementation showed a positive outcome in both aerobic and anaerobic performance, which was related to physical test outcomes. The authors also learned that the best time period to take an Arg supplement was 60-90 min before exercise, and the most effective dosage corresponded with the athlete’s body weight (0.15 g/kg).
Although the authors should be commended for their work, this reviewer has some questions, comments, and points of clarification regarding the reporting of this manuscript that need to be addressed before considering the manuscript further for publication consideration. To help guide and direct the author’s efforts, the reviewer has raised each question, point of clarification or concern point-by-point below.
Methods:
REVIEWER: Section 2.3: It seems that only one rater screened and selected the studies for inclusion. According to the AMSTAR 2 checklist, for best methodological quality, systematic reviews and meta-analyses should include at least two screeners who should independently screen and agree upon the selected studies for eligibility. Alternatively, the two reviewers should each select a sample of the same eligible studies and achieve at least 80% agreement between those studies with the remainder of the studies being screened by one reviewer. This reviewer therefore requests that the authors please clarify whether or not the reviewers of the submitted manuscript adhered to this protocol or used a different protocol.
AUTHORS: Thank you for your help. In order to avoid misunderstandings, the authors have remade section 2.3 to indicate that 2 authors carried out the search, selection of articles, as well as the determination of risk of bias in parallel (independently of each other). Subsequently, all the findings were pooled, obtaining more than 90% agreement between authors.
REVIEWER: Section 2.4: Please include a more in-depth description of the steps taken to perform the statistical analysis of the performance outcome variables, specifically VO2 max.
AUTHORS: Thank you for your suggestion. The authors have included some information about analysis of the performance outcome variables, specifically VO2 max.
Concretely, the outcomes obtained by Repeated Sprint Ability Test (RSAT), strength exercises as isokinetic flexion, isokinetic extension and bench press, 1 min all out test, 1km time trial (TT) and Wingate Test were included in > VO2max. On the other hand, the outcomes obtained by Incremental test to exhaustion, 2x5km TT, 60 min test at 80% of ventilatory threshold (VT), a 16.1km TT and 2x6min running followed by a running test until exhaustion. Hardvard Step Test to measure VO2 max. capacity were included in VO2 max. For the statistical analysis, the sample sizes, means and standard deviations of the different outcomes studied were extracted both in the group supplemented with Arg and in the control group and in the pre and post treatment. When there was no numerical data, it was requested to the authors or if the data were plotted as figures, the values were estimated based on the pixel count using images calibrated in ImageJ software (National Institutes of Health, Bethesda, MD, EE).
REVIEWER: Please provide a more comprehensive discussion of the methods used to code the 15 articles included in the analysis and how inter-rater reliability and validity were established with Kappa and Pearson’s r statistical procedures.
AUTHORS: Thank you for your recommendation. The authors have included some information about inter-rater reliability and validity in different points in methods section. In this sense, the authors have performed the inter-rater reliability and validity by Cohen´s kappa coefficient.
Results
REVIEWER: Section 3.5/3.6: Please include a calculation of the Hedges’ g effect size to improve clarity of data presented. Please also provide the established reliability statistics.
AUTHORS: Thank you for your suggestion. The authors have modified these sections to improve clarity of data presented.
Figure 4 shows that Arg presented small and significant effect on anaerobic performance (Hedges’ g (SMD) =0.24; 95%CI (0.05 - 0.43); p = 0.01). The meta-analysis reported small amount of inconsistency between studies reviewed (I2= 0%; p = 0.85).
Arg showed large and significant effect on aerobic performance (Hedges’ g (SMD) = 0.84; 95%CI (0.12 - 1.56); p = 0.02) (Figure 5). The meta-analysis reported large heterogeneity between studies reviewed (I2= 89%; p < 0.001).
REVIEWER: Please update and provide a clearer description for table 3.
AUTHORS: Thank you for your appreciation. The authors have updated and have provided the information regarding Table 3.
Other comments:
REVIEWER: Please consider revising the submitted manuscript with AMSTAR 2 guidelines in consideration of improving the overall clarity and transparency of the manuscript with regard to the reported study findings.
AUTHORS: Thank you for your suggestion. The authors have included in the methods section some information to fill AMSTAR 2 checklist. Moreover, authors have calculated the quality of this systematic review and meta-analysis and have included this information in the Strength, Limitations and Future Lines of Research: “In fact, when assessing the review and meta-analysis quality by AMSTAR guidelines, the score obtained was “high quality” [74].
REVIEWER: Please include a description of any publication bias procedures that were performed for the current study and state whether publication bias was existent/non-existent.
AUTHORS: Thank you for your help. The authors have added this information in the 2.5. Publication bias section. Moreover. The authors have some information about this in 4.2 and 4.3 sections.
REVIEWER: Please double check and correct any typographical errors throughout the manuscript.
AUTHORS: Thank you for your recommendation. The authors have checked some typographical errors throughout the manuscript

Reviewer 2 Report
Major
Overall, this is good work. Though, I feel it necessary to “clean up” the introduction a bit, as it is over-informing beyond the needs of the introduction then repeated in the discussion. It should suffice to say something along the lines of, “x number of studies found positive effects and n number of studies found no effect of arginine supplementation and these investigations used varying methods regarding the dose and timing. Therefore, a meta-analysis and systematic review is necessary to identify the conditions which arginine supplementation may or may not yield benefit,” instead of discussing the studies in detail. As is, it’s giving away the ending before the problem and study are presented.
Throughout, be sure to change from a comma to a decimal to indicate fractions. e.g., 1,5g/kg à 1.5g/kg.
Minor
n 71 p2
p3 n 104 – author’s à authors’
p4 ln 158 – established
p6 ln 209 – suggestion, remove “and confusing”
p6 ln 201 – suggestion, a total of 15 were included for analysis.
p7 ln 219 – fight à combat
p8 ln 231 – delete “a” before chronic
p17 ln 347 – ON à NO?
Author Response
Point-by-Point Response to Reviewer’s Comments
We would like to sincerely thank the reviewers for their helpful recommendations. We have seriously considered all the comments and carefully revised the manuscript accordingly. Revisions are highlighted in yellow through the manuscript to indicate where changes have taken place. We feel that the quality of the manuscript has been significantly improved with these modifications and improvements based on the reviewers’ suggestions and comments. We hope our revision will lead to an acceptance of our manuscript for publication in Nutrients.
In advance,
King regards
Reviewer 2:
REVIEWER: Though, I feel it necessary to “clean up” the introduction a bit, as it is over-informing beyond the needs of the introduction then repeated in the discussion.
AUTHORS: Thank you for your suggestion. The authors have cleaned up the introduction as far as possible, in order to identify the conditions which arginine supplementation may or may not yield benefits and avoid specific discussion.
REVIEWER: Throughout, be sure to change from a comma to a decimal to indicate fractions. e.g., 1,5g/kg à 1.5g/kg.
AUTHORS: Thank you for your observation. The authors have changed all commas indicating fractions throughout the manuscript.
REVIEWER: Author´s ln 104.
AUTHORS: Thank you for your indication. The authors have modified the error.
REVIEWER: Established ln 158.
AUTHORS: Thank you for your correction. The authors have corrected the word.
REVIEWER: Remove “and confusing”.
AUTHORS: Thank you for your observation. The authors have removed it.
REVIEWER: Suggested, “a total of 15 were included for analysis”.
AUTHORS: Thank you for your suggestion. The authors have modified the sentence.
REVIEWER: Change fight for combat.
AUTHORS: Thank you for your indication. The authors have changed this word.
REVIEWER: Delete “a” before chronic.
AUTHORS: Thank you for your observation. The authors have modified it.
REVIEWER: Change ON à NO.
AUTHORS: Thank you for your indication. The authors have changed this error.

Reviewer 3 Report
The authors present a well done and needed study regarding the supplementation of Arg in athletic performance. The methods and conclusions are appropriate and this reviewer has no further comments. My compliments to the authors!
Author Response
Reviewer 3
REVIEWER: The authors present a well done and needed study regarding the supplementation of Arg in athletic performance. The methods and conclusions are appropriate and this reviewer has no further comments. My compliments to the authors!
AUTHORS: Thank you very much for reviewing our manuscript. Your comments give strength to continue working in this way.
Kind regards,
Round 2
Reviewer 1 Report
The reviewer thanks the authors for addressing the items mentioned in the first review. After careful consideration, the reviewer is comfortable moving forward and recommending the paper for further publication consideration.